# The One, the Many and Koinonia: Synodality and Receptive Ecumenism

**N. Ammon Smith**

Glenmary Home Missioners, Cincinnati, OH 45246-5618, USA; nsmith@glenmary.org

**Abstract:** This essay will explore the relationship between synodality as an ecclesiality and Receptive Ecumenism as one form of enacting the synodal vision within the ministry of ecumenism. In so doing, this essay will consider how Receptive Ecumenism within the ministry of Christian unity fosters transformation and *koinonia* within and between communions and presents an ecclesial vision for the People of God that is analogous to the Trinitarian *koinonia*, thus offering a vision for ecumenism that avoids both Parmenidean homogeneity or Heraclitian flux, which has plagued the ecumenical endeavor.

**Keywords:** synodality; ecumenism; Receptive Ecumenism; koinonia

## 1. Introduction

In his Bampton Lectures, given in 1992, Colin Gunton introduced a philosophical quandary, which he describes as the issue of the "one and the many". Utilizing an ancient debate between Heraclitus and Parmenides as a heuristic device, Gunton sets the scene for what he sees as a question salient to contemporary thought. Though, admittedly, the sources tend to avoid providing exact details of this debate, Gunton posits that Heraclitus' position can be understood as seeing that "everything is flux, and that war is the universal creative and ruling force, reality being suffused by forces pulling in both ways at once, so that the basic fact in the natural world is strife" (Gunton 2005, p. 16). This is not to suggest that order is inconsequential or unreal, but that it exists within plurality and motion, for the many precede the one. The only constant is change, one might say. Contrastingly, Parmenides suggests that the real is "totally unchanging" (Gunton 2005, p. 18). In this way, "reality is timelessly and uniformly what it is . . . The many do not really exist, except it be as functions of the One" (Gunton 2005, p. 18). Gunton notes that in the past, philosophers, such as Coleridge, posit that Greek thought tended towards a monism and that today, within our so-called postmodern era, one may assume a more Heraclitean tendency, with its recognition of the role of context and truth being situated therein. This assumption Gunton would question, suggesting instead that the dialectic between the one and the many tends to collapse one into the other. This argument, applied by Gunton to the broader societal sphere, this essay will suggest, can also be applied to the broader ecumenical movement. Thus, in order to avoid this constant vacillation, a consideration of a third way, which Gunton proposes, may supply new horizons for ecumenism as well. This Gunton refers to as the "Three", or *koinonia*.

Following Gunton's argument, this essay will begin by describing briefly the state of the contemporary ecumenical movement as it exists within the West. This will include, as will be shown, a tendency towards either the Parmenidian or Heraclitean schemas, which has contributed to the ecumenical movement being in a kind of "cul-de-sac", with little momentum towards any form of structural and sacramental unity between the churches. Following this setting of the scene, this essay will introduce the synodal emphasis taken by the Catholic Church in recent years in order to invite the Church to become a "synodal church". This is described as a listening church where the "specific *modus vivendi et*

*operandi* of the Church, the People of God, which reveals and gives substance to her being as communion when all her members journey together, gather in assembly and take an active part in her evangelizing mission" (International Theological Commission 2018, no. 6). This synodal effort, as can be seen within the 2018 document from the International Theological Commission, *Synodality in the Life and Mission of the Church*, provides due attention to the manner in which all the baptized, even those coming from communions outside of the Catholic Church, are invited into the synodal process. Finally, this essay will end by introducing the natural question as to how synodality encourages and energizes the ecumenical movement. Said another way, how does the synodal movement not simply throw the ecumenical movement from one side to the other of the ancient debate, between the one and the many, highlighted by Gunton? In an attempt to address this concern, this essay will end by providing an argument for the utilization of Receptive Ecumenism as a *modus vivendi et operandi* of the synodal church. Receptive Ecumenism, being somewhat new to the ecumenical scene, has generated much interest due to its encouragement towards a disposition of listening and learning. It will also be argued that when viewed through the lens of *koinonia*, Gunton's own answer to the vacillating dialectic, Receptive Ecumenism provides a means of fostering transformation which leads to deepening communion through the reception of gifts—to use the language of John Paul II—and the need for each communion to ask where this reception and conversion may take place first within their own tradition.

## 2. Setting the Stage: The One and the Many of the Modern Ecumenical Movement

The modern ecumenical movement, which many understand as beginning at the start of the 20th century, can be seen as an engagement with the question of the one and the many. The flurry of missionary efforts taking place throughout the world for a number of centuries had allowed for a propagation of the Christian faith on an enormous scale. This increase led to the helpful, albeit difficult, realization of the multitudinous wounds existing within the body of Christ, that being the various divisions separating Christian communions. In his essay on African perspectives of Receptive Ecumenism Agbonkhianmeghe Orobator highlights the realities and difficulties that come from these divisions. Sharing a response to the competing missional efforts of both the English Baptists and the French Catholics in the 19th century, Orobator notes the clever response from King Pedro V of the former Kongo Kingdom in Central Africa, "'You white men, you perplex me with your different teachings,' complained the King. 'I don't know how to choose between you . . . I shall keep both these palavers in my heart, and when I appear before God, He must decide and judge both'" (Orobator 2022, pp. 241–42). This reality came to much clearer view in 1910 at the World Missionary Conference in Edinburgh as missionaries from around the globe met to compare notes and share stories. As this sharing commenced a deeper realization of the harm that the divisions between communions had caused, thus confusing the newly baptized and creating new divisions among peoples. This can be seen as a kind of hyper-Heraclitean schema taking hold and creating unhealthy flux in the members of the body of Christ.

From this impetus for deepening unity among Christians would come the familiar two-fold approach towards ecumenical rapprochement, that being the Life and Works and Faith and Order movements. The former, founded in Stockholm in 1925 with the rather unhelpful slogan, "Doctrine divides, service unites" (Braaten and Jenson 2003, p. 21), tends towards organizing various communions and local churches in the ongoing need to care for the poor along with a growing recognition of and attention to the ecological crisis. Around this same time the Faith and Order movement was formed, which sought to engage issues of doctrinal difference and division. Early on in these dialogues, attention was given to topics highlighted within the Chicago–Lambeth Quadrilateral. This proposal towards Christian unity centered around the canonical scriptures, the ancient creeds, an emphasis on the sacraments of baptism and the Lord's Supper, and the historic episcopate "locally adapted" (Braaten and Jenson 2003, p. 20). This, along with the publishing in 1920 of the encyclical,

*Unto All the Churches of Christ Wheresoever They Be*, from the Orthodox Patriarchate of Constantinople, calling for a "fellowship of churches" to foster rapprochement through study and cooperation, saw the beginning of numerous bi-lateral dialogues focused on the topic of doctrinal unity.

These areas of ecumenical focus have provided the opportunity for the fostering of significant partnerships in ministry for nearly a century and have helped to produce numerous theological insights and ecumenical convergences, such as *Baptism, Eucharist and Ministry* (1982) and the *Joint Declaration on the Doctrine of Justification* (1999), that would not have been anticipated even a century earlier. With the entrance of the Catholic Church into the formal ecumenical scene with Vatican II, and in an important preliminary fashion with the forming of the *Catholic Conference for Ecumenical Questions*, which included theologians such as Yves Congar and Karl Rahner, many considered that a reunification of the various communions to be immanent. This somewhat Parmenidian view began to lose steam towards the end of the 20th century, even with the incredible gains seen through the signing of the *Joint Declaration* and the helpful reiteration of Catholicism's commitment to ecumenism given that same decade. I say "somewhat Parmenidian" as Vatican II saw the end of any formal emphasis on an ecumenism of return of Protestant groups into the Catholic Church, which could be understood as a hyper-Parmenidianism. Yet, for many in the final third of the 20th century, the belief in an immanent structural and sacramental union between the Catholic Church and various other churches was real and should not be seen as overly optimistic or naïve.

This expectation changed overtime with both the growth of new questions, particularly around the practice of ordaining women within many Protestant churches, diverging ethical views between communions, along with a general frustration due to the lack of movement occurring between the churches towards any kind of structural or sacramental unity. In response, some within the ecumenical movement turned their focus to a Life and Works ecumenism. This can be seen within the World Council of Churches, under the leadership of Konrad Raiser, which created a model of ecclesial connection that is "structurally distinct yet fraternally associated" by choosing to focus much of their own attention away from issues of doctrine and order, to "mission and evangelization" (Murray 2008, p. 10). This, mixed with a growing interest in and recognition of the need for interreligious dialogue, began to further push questions of doctrine to the periphery. In an incredible statement from the signers of the *Princeton Proposal for Christian Unity*, this change was undeniable: "It is now generally agreed that the classical interests of Faith and Order... have been marginalized" within the World Council of Churches (Braaten and Jenson 2003, p. 25). From here, a somewhat Heraclitean view comes to the fore with some Protestant denominations experimenting with various partnership structures based on their shared aims. This can be seen within many "church network" models, which allow for churches from differing denominations to join their group, as well as the rather novel phenomenon of individual churches choosing to be a part of more than one denomination at the same time. For others, the question of structural unity is now spoken of as a bygone era, citing the explosive growth of the nondenominational movement as evidence. Yet, as the Catholic theologian Paul Murray has pointed out, perhaps this fever pitch of the ecumenical movement ran ahead with such force and has now lost traction not because anyone had a clearer view of what unity actually looks like, but because what was removed initially was the "softwood" of our divisions, leaving now the "hardwood" for the various Christian communions to engage with (Murray 2014, p. 3).

Murray provides various examples of these "hardwoods", such as the differences in view regarding the relationship between the local and the universal Church, divergent understandings of who can be legitimately ordained, as well as competing understandings of the manner of church governance and decision making. When attempting to address all of these questions, not to mention the many covered throughout the latter half of the past century, one may become overwhelmed by the differences and the work required to reconcile them. Yet, the ecumenical question, though including questions of authority

and justification, at root is a question of ecclesiology; that is a question of the nature of the Church. This reality has become the focal point of much contemporary ecumenical theology and was a seminal question at the Second Vatican Council as well.

### 3. Koinonia as a Vision for the Church

Recognizing the connection between ecumenism and ecclesiology, it is perhaps not surprising that as the growth in interest in formal ecumenical engagement began to grow, particularly within Catholic circles, conversation regarding the nature of the Church also began to develop in a substantial way. An example of this interconnection between ecclesiology and ecumenism can be seen in the very fact that both the Decree on Ecumenism and the Dogmatic Constitution on the Church were promulgated at the Second Vatican Council on the same day. One of the more influential theologians related to both of these topics was the Dominican, Yves Congar, who, as noted by Benoit-Dominique de La Soujeole, was the first to intuit *koinonia* as a definition of the Church, thus providing a framework for the Church as communion (Soujeole 2014, p. 451).

Pulling on the New Testament data, *koinonia* can be seen within three distinct meanings which, according to Soujeole, "express three aspects of the same reality" (Soujeole 2014, p. 455). These meanings are understood as "contribution, invitation" and "sharing", as seen within the encouragements from the apostle Paul for others to share with the poor Christians in Jerusalem (2 Cor. 9:13, Rom. 15:26; see also *The Church: Towards a Common Vision* no. 13). *Koinonia* is also translated as a "participation" and a "taking part", which is again used by Paul to describe one's participating in the Eucharistic feast (1 Cor. 10:16), as well as the sufferings of Christ (Phil. 3:10). Finally, *koinonia* is described as "community", or fellowship, as seen within the first epistle of John when describing a Christian's fellowship with the Father and the Son and His Church (1 Jn. 1:3, 6–7). Thus, Soujeole sums up that the use of the term *koinonia* is a "question of expressing three distinct but related meanings: to communicate—to partake—to be/have in common" (Soujeole 2014, p. 462).

Looking to the patristic data it is significant to note that Irenaeus, in his works to combat heresy and to foster the unity of the Church, utilizes the term *koinonia* more than eighty times. Key to the Gnostic debate was the belief that "communication between God and creation, eternity and time", was an impossibility (Soujeole 2014, p. 456). The contrast between this cosmology and the one presented by the Incarnate Christ are clear. For the bishop of Lyon the mission of God in the incarnation—the divine plan of salvation—is the restoration of humanity with God and to communion. This is brought about firstly by the Trinitarian God, who exists in *koinonia*, extending grace to creation in the Incarnation, taking on human nature which had lost access to *koinonia* with God, "The Word has saved that which really was [created, viz.] humanity which had perished, effecting by means of Himself that *koinonia* which should be held with it, and seeking out its salvation" (Soujeole 2014, p. 456, quoting *Adversus Haereses* V.14.2). Thus, the Trinitarian *koinonia* is extended to humanity in the Incarnation. This, as Soujeole rightly remarks, is the extension of grace, something he identifies as Christic. This emphasis on the Christic nature of grace is because that which is offered to humanity is an invitation to communion with God, or *koinonia*, which provides a unique quality to the good offered. To this, and with Paul, we might say, "Christ in you, the hope of glory" (Col. 1:27). It is also in this extension of Christic grace that the *koinonia* of the Church is formed, as all are called to participate in *koinonia* with God, and this occurs, by its communal nature, within the Church. In this way, the Church also receives its vocation to extend this *koinonia*, through the Holy Spirit, to the whole world. This calls for a deepening of unity, of sharing, participating, or *koinonia*, between the members of the Church, so as to more rightly reflect the *koinonia* of the Trinity. In this way, the image of the Church as *koinonia* has begun to shape contemporary ecclesial understandings and provide a connected path forward between ecclesiology and ecumenism.

## 4. Koinonia and the Synodal Church

The biblical and patristic emphasis on the Church as *koinonia* has recently found a uniquely catalyzing force within the Catholic Church's global synod process. As noted previously, the goal of this worldwide synodal approach is to foster a "synodal church", where each member is taking an active part in journeying with the ecclesial whole towards its evangelizing mission. In this way, the synodal process is not simply a kind of survey to find what others think about any number of issues. Instead, it is the introduction of what might be called a *habitus*, a way of being, in order to foster the kind of church that is synodal.

Speaking of the manner in which the Catholic Church may be understood functionally as synodal, the aforementioned document *Synodality in the Life and Mission of the Church* introduces two key concepts. First is what is referred to as the "all", the "some", and the "one". Pulling on the doctrine of the *sensus fidei* along with the "sacramental collegiality of the episcopate" in communion with the Bishop of Rome, synodality can be seen as involving all the baptized (all), along with the ministry of the college of bishops (some) and the unique role of unity which is exercised by the ministry of the Roman Pontiff (one). In this way, "The dynamic of synodality thus joins the communitarian aspect which includes the whole People of God, the collegial dimension that is part of the exercise of episcopal ministry, and the primatial ministry of the Bishop of Rome". (International Theological Commission 2018, no. 64). The document goes on to note that it is in this journeying together, the faithful and their pastors together, that a "*singularis conspiratio*", which is an icon of the eternal *conspiratio* that is found within the life of the Trinity, is promoted and expressed ecclesially. Another important concept is the framing of this synodal Church within Pope Francis' image of an "inverted pyramid", which includes the "all", the "some", and the "one", but with the "summit below the base", following Christ's own example of ministers being the "least of all" (International Theological Commission 2018, no. 57). In this way, a hyper-Parmenidian–Heraclitean dichotomy gives way to the mutual sharing and journeying that is indicative of the *koinonia* expressed within the life of the Trinity and is representative of the nature of the Church, which is the body of Christ.

This image of the Church as synodal connects with the icon of the *koinonia* of the Trinity as well as emphasizing the need for the Church to be "constantly moving forward toward the fullness of divine truth" (International Theological Commission 2018, no. 64), as well as "fac[ing] up to the question: how can we truly be a synodal Church unless we live 'moving outwards' towards everyone in order to go together towards God?" (International Theological Commission 2018, no. 109). This twofold emphasis of *koinonia* and mission can be seen as the two-step cadence of Christ's prayer for the unity of his disciples so the world may know that the Son has been sent from the Father (Jn 17:21). Thus, a synodal Church, that being a listening and missioning Church, is also one committed to the deepening of unity among all the baptized. In this way, it can be seen that synodality is at the "heart of the ecumenical commitment of Christians" and "represents an invitation to walk together on the path towards full communion and because—when it is understood correctly—it offers a way of understanding and experiencing the Church where legitimate differences find room in the logic of a reciprocal exchange of gifts in the light of truth" (International Theological Commission 2018, no. 9). Therefore, a synodal Church, which is at its core ecumenical, invites a legitimate exchange of gifts, following the language of John Paul II, and allows room for difference in the light of the Word (International Theological Commission 2018, no. 109). Here, the ecumenical emphasis on unity, not uniformity, is affirmed.

This emphasis on the connection between synodality, *koinonia*, and ecumenism is highlighted through the International Theological Commission's reference to the World Council of Churches' convergence document, *The Church: Towards a Common Vision (TCTCV)*. Here, *TCTCV* makes explicit this connection when it states, "under the guidance of the Holy Spirit, the whole Church is synodal/conciliar, at all levels of ecclesial life: local, regional and universal. The quality of synodality or conciliarity reflects the mystery of the trinitarian life of God, and the structures of the Church express this quality so as to actualize the

community's life as a communion" ([World Council of Churches, Commission on Faith and Order 2013](#), p. 53). This actualizing of the communion of the Church, being guided by the Holy Spirit, is the mutual sharing and participating, at all levels, that makes up the synodal act as it lives out the *koinonia* seen within the Trinitarian life. This is not to suggest that synodality and *koinonia* are synonymous. Instead, it is to note the former's unique role, which is the act by which the ecclesial body attunes itself to the Spirit in the mutual listening and sharing, and how this reflects *koinonia*, which is indicative of the "mystery of the trinitarian life of God". Additionally, *TCTCV* notes that "Communion, whose source is the very life of the Holy Trinity, is both the gift by which the Church lives and, at the same time, the gift that God calls the Church to offer to a wounded and divided humanity in hope of reconciliation and healing" ([World Council of Churches, Commission on Faith and Order 2013](#), p. 1). The connections between the Catholic Church's understanding of synodality and the call to the living out of the Church as *koinonia* through *TCTCV* could not be clearer. Yet, it may be asked, for all the language of "communion, participation, fellowship [and] sharing" ([World Council of Churches, Commission on Faith and Order 2013](#), p. 13), what does this exchange look like ecumenically? Said differently, what does it look like for communions living now as the wounded body of Christ, with her manifest divisions, to begin to practice synodality and *koinonia* with other Christians so as to actualize the unity that Christ prayed for? To answer this question this essay will suggest the practice of Receptive Ecumenism, though of course others could be included as well. Also, to avoid any further abstraction of what should be a lived reality, that being the synodal Church, this essay will seek to describe Receptive Ecumenism by providing examples of this expression of *koinonia*.

## 5. Receptive Ecumenism and Synodality

In the Spring of 2022, I and several colleagues hosted a Global Christian Forum-style gathering on the campus of Pentecostal Theological Seminary. The Forum seeks to foster a deepening of unity among Christians from varying backgrounds by simply inviting them to share their own experience of Christ, another expression of synodality and *koinonia* expressed ecumenically. This event, held over two days, brought together the local Catholic diocese, members of Glenmary Home Missioners, a Catholic society of apostolic life, and three different Pentecostal denominations. As the organizers of this event, we felt it important to encourage a deepening of connection among these Christians by sharing with them the principles of Receptive Ecumenism as well as providing plenary speakers from both traditions to share on their own experience of Receptive Ecumenism.

The Catholic representative for this plenary was a Franciscan friar, associated with the ecumenically inclined and pioneering community of the Franciscan Friars of the Atonement. As requested, this friar, who spoke in the Pentecostal chapel while wearing his traditional Franciscan habit, spoke briefly about the Franciscan tradition and how his own dress, particularly the knots in the cincture around his waist, reminds him of his call to the Evangelical Counsels of poverty, chastity, and obedience. It was this artifact, worn day in and day out, that reminded him of his commitments and, he noted, brought him a sense of freedom and joy. In order to facilitate sharing among the large group of attendees, we organized small groups with both Catholic and Pentecostal members. During this, our second day together, we posed the question to these groups, "What did you hear within these plenaries and from a tradition outside your own which elicited a sense of need or desire for you and for your own communion?" In response to the habited friar, one Pentecostal pastor shared with my table that for much of his life he understood traditional Catholic attire, including the Franciscan habit, to be a form of legalism that Christ's death and resurrection had overcome. I am unaware of any definitive statement on this belief. Nor do I believe this to be any formal ecclesial position which would separate communions. Still, for this pastor, this division was real and manifest in Catholic garments.

The pastor continued that, after hearing this friar share how his habit provided him a kind of freedom, he himself desired a similar artifact that he might take up to remind

him of the freedom he has in Christ. It was in this simple experience of sharing between traditions and among individuals from very different ecclesial realities that a sense of desire, perhaps even need, was brought to the surface. Though our time did not allow for a continued investigation of the manner in which a Pentecostal may learn from and perhaps even receive gifts from this Franciscan, it was this initial feeling, this sense of need or desire, which catalyzes the practice of Receptive Ecumenism.

Developed by the Catholic theologian Paul Murray, Receptive Ecumenism is "concerned to place at the forefront of the Christian ecumenical agenda the self-critical question, 'What, in any given situation, can one's own tradition appropriately learn with integrity from other traditions?'" (Murray 2008, p. 12). Much formal ecumenical dialogue seeks to define with further clarity a tradition's own doctrinal views on any number of topics, but Receptive Ecumenism invites each tradition into a practice, even a *habitus* or a "total ethic" (Murray 2008, p. 16), of receptive learning with "dynamic integrity" by asking questions about the other and to consider what they might receive. The theological lynchpin of Murray's argument is developed from the Catholic Church's own ecclesial understanding of its relation to the Church of Christ and the manifold expressions of these ecclesial elements, what are oftentimes referred to as "gifts", manifest within other communions (Second Vatican Council 1964, *Lumen Gentium*, no. 8); (Second Vatican Council 1965, *Unitatis Redintegratio*, no. 3). This ecclesial vision, which includes implicitly a view towards *koinonia* with its emphasis on listening, sharing and communing together, further legitimizes the opportunity for, and need of, the Catholic Church to enter into a learning venture as the pilgrim people of God. This commitment to Catholic learning, Murray notes, can be seen within the Church's own emphasis on the need for a continual purification, *semper purificanda*, and when directed ecumenically looks like the reception of gifts, or expressions of the Church of Christ found within another communion, to and for our tradition's needs and continual growth. Turning once again to the aforementioned Pentecostal minister, it was his experience of the Church of Christ found within a tradition outside his own that the opportunity for learning began to present itself.

Implicit within this encounter, as has already been stated, is the recognition of one's need or desire catalyzing this opportunity for the reception of gifts from another. Yet in connecting this sharing to expressions of the Church of Christ, and not simply appropriating what is believed to be a good or expedient idea from another tradition, there is an immediate connection to the Holy Spirit in both nurturing this sense of need or desire as well as "inspirating" one's movement towards this expression of the Church of Christ as located within another ecclesial tradition (Murray 2018). The reception of these expressions are performed with what Murray calls "dynamic integrity", that is the integration of the gifts of another into one's own tradition not in a like-to-like manner necessarily, but in such a way that that which is received hangs coherently with the other commitments, theological and lived, of the learning communion.

Another example of Receptive Ecumenism took place at a gathering of Christian leaders, hosted by the Centre for Catholic Studies at Durham University in England, and on the topic of synodality. This gathering included members of various denominations including the Church of England, the United Reformed Church, Pentecostals, Quakers, Methodists, Baptists, and the Catholic Church. The guiding question of these meetings was the following: what elements and in which contexts might the Catholic Church learn from another's ecclesial expression of synodality? Participants were organized within various groups, which included members from the aforementioned ecclesial communions, along with Catholic facilitators, scribes, and a designated canon lawyer and theologian. Conversations centered around prepared documents seeking to answer these two questions and included times of prayer, singing, listening, and silence. Following the day's meetings, these Catholic members met to discuss their findings and to consider together what areas of learning and reception, within dynamic integrity, might be had from the various ecclesial traditions' own experience and practice of synodality.

As shown from the above examples, the practice of Receptive Ecumenism aligns itself with the synodal emphasis of the "pilgrim character of the Church" (International Theological Commission 2018, no. 49). The reception of gifts is less a methodology and more, as Murray puts it, a "way", which is a "process of growth and change—a process of conversion—that is at root not a loss, nor a diminishment but a finding, a freeing, an intensification, and an enrichment" (Murray 2008, p. 6). As communions walk together, sharing and participating, in the life of the other, new opportunities for receptive learning and synodal engagement emerge. Here the concept of a "thick ecumenism", as described by Baptist theologian Steve Harmon (Harmon 2006, p. 16), begins to take shape as an ecumenism that does not look for the least common denominator. Instead, thick ecumenism recognizes the Spirit-given abundance of another and, thus, the opportunity for sharing with others to foster a conversion, that is, a further becoming of the learning, listening, and growing Church into the fullness of Christ who fills all in all (Eph. 1:23).

## 6. Conclusions

At the conclusion of Gunton's lectures, cited at the beginning of this essay, he turns to the notion of *koinonia* as a way of responding to the Parmenidian and Heraclitean dichotomies. As this essay has argued, and as is expressed within the vision of the synodal Church as well as Gunton's own research, it is the understanding of *koinonia* which is most proper to the being and nature of the Church. Yet, as Gunton notes, too often the Church has operated as an institution, which at times limits dialogue and sharing (see Third Anglican–Roman Catholic International Commission 2018, pp. 95–96), and, at its most de-natured, operates out of a sense of lack or scarcity. The synodal vision of the Church provides a different vision in which the sharing of gifts, John Paul II's famed description of ecumenism, is not a complete removal of oneself or loss, as might be seen within some postmodern views of "pure gift". Instead, a synodal Church, engaging in the ministry of ecumenism, is a filling up, an abundance, and a sense of becoming more and more into the image and likeness of Christ. It is within this ecclesial vision that the practice of Receptive Ecumenism provides a way for the Church's continual becoming, becoming more like ecumenical others, but also becoming more themselves, i.e., the Body of Christ, by receiving these expressions of the Church of Christ manifest. In this sharing, there is also a recognition of the "legitimate plurality", which is indicative of the Body of Christ whose Head has brought salvation in both "heaven and earth" (Eph. 1:10). In this way, the hyper-Parmenidian and hyper-Heraclitean duality, the competing visions of the one and the many, give way to the *koinonia* of the pilgriming Church, whose mission it is to proclaim the God who is love within the Trinitarian *koinonia*.

**Funding:** This research received no external funding.

**Institutional Review Board Statement:** Not applicable.

**Informed Consent Statement:** Not applicable.

**Data Availability Statement:** Not applicable.

**Conflicts of Interest:** The author declares no conflict of interest.

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
