# Peer review of "The One, the Many and Koinonia: Synodality and Receptive Ecumenism"

_religions, doi:10.3390/rel14111393_

Round 1

Reviewer 1 Report

Comments and Suggestions for Authors

Comments on the Quality of English Language

 Theere are minor English language editting which will be the Journal Language editor.

Author Response

Thank you. I have made the suggested grammatical edits.

Reviewer 2 Report

Comments and Suggestions for Authors

I find the basic idea employed in this article interesting and important. The author combines the two cognitive models: Receptive Ecumenism and the dichotomy of Heraclitian and Parmedidean paradigms, and this combination is fresh and contributes to the fostering of ecumenical knowledge. Hence, I recommend the article for publishing. 

However, I want to pay attention to two areas for improvement. First, when describing the failures and difficulties of the ecumenical movement at the beginning of the 21st century, the author did not refer to the divisions in ethical teaching which tore apart the Churches. Second, when referring to receptive ecumenism, there is a lack of one or two paragraphs that would systematically describe the theoretical definition of receptive ecumenism. 

Author Response

Thank you for your review.

Your point about diverging ethical views is well taken and I have added a brief line in my essay about that. I have also added another section which provides another example of Receptive Ecumenism which, I hope, gives better clarity to its methodology.

Reviewer 3 Report

Comments and Suggestions for Authors

Author Response

Thank you for your review. This is really helpful to me.

I have made some adjustments:

  1. I incorrectly included the abstract within the body of the paper so I have removed this redundancy, which should also clarify the overall goals of the essay.
  2. I have expanded on the section on koinonia so as to give it greater definition.
  3. Receptive Ecumenism is now listed in the abstract and within the introduction section. I have also expanded upon it towards the end of the essay.